# Brain Control Reproduction by the Endocrine System of Female Blue Gourami (*Trichogaster trichopterus*)

**DOI:** 10.3390/biology9050109

**Published:** 2020-05-21

**Authors:** Gad Degani

**Affiliations:** 1MIGAL–Galilee Research Institute, POB 831, Kiryat Shmona 1101602, Israel; gad@migal.org.il; 2Faculty of Sciences, Tel-Hai Academic College, Upper Galilee 1220800, Israel

**Keywords:** Anabantidae, hormone receptors, *Trichogaster*, vitellogenesis, yolk accumulation, gonadotropic brain pituitary gonad axis

## Abstract

Blue gourami belongs to the Labyrinithici fish and the Anabantiform order. It is characterized by a specific organ located above its gills for the respiration of atmospheric oxygen. This specific adaptation to low oxygen levels affects reproduction that is controlled by the brain, which integrates different effects on reproduction mainly through two axes—the gonadotropic brain pituitary gonad axis (BPG) and the hypothalamic-pituitary-somatotropic axis (HPS axis), including the interactions between them. This brain control reproduction of the Anabantoidei suborder summarizes information that has been published on the hormones involved in controlling the reproduction system of a model female blue gourami fish (Trichogaster trichopterus), including unpublished data. In the whole-brain transcriptome of blue gourami, 17 transcription genes change during vitellogenesis in the brain. The hormones involved in reproduction in blue gourami described in the present paper include: Kisspeptin 2 (Kiss 2) and its receptors 1 and 2 (KissR 1 and 2); gonadotropin-releasing hormone 1, 2 and 3 (GnRH1, 2 and 3); GnRH receptor; pituitary adenylate cyclase-activating polypeptide (PACAP) and its related peptide (PRP); somatolactin (SL); follicle-stimulating hormone (FSH); luteinizing hormone (LH); growth hormone (GH); prolactin (PRL), 17β-estradiol (E2); testosterone (T); vitellogenesis (VTL); and 17α,20β- dihydroxy-4-pregnen-3-one (17,20P). A proposed quality model is presented regarding the brain control oogenesis in blue gourami that has a Labyrinth organ about which relatively little information has been published. This paper summarizes the complex various factors involved in the interactions between external and internal elements affecting the brain of fish reproduction in the Anabantiform order. It is suggested to study in the future the involvement of receptors of hormones, pheromones, and genome changes in various organs belonging to the reproduction system during the reproduction cycles about which little is known.

## 1. Introduction

Blue gourami (*Trichogaster trichopterus*) belongs to the Labyrinithici fish and the Anabantiform order. It is characterized by a specific organ located above its gills for the respiration of atmospheric oxygen (Labyrinth) [1]. The geographic distribution of the suborder of Anabantoidei fish is Central Africa, India and southern Asia [2]. In natural habitats, the Anabantoidei fish blue gourami (35) adapts to unpredictable habitats in which water oxygen concentration varies throughout the year and reaches a very low percentage according to labyrinth organ development [1].

Labyrinth fish undergo two different periods during their life cycle: before labyrinth organ development from eggs to juveniles and the retention of oxygen over the entire surface by diffusion; and after organ development when it becomes important for the organ to breathe [1,3,4]. The adaptation by the eggs and fry to develop in water with low oxygen concentrations involves laying eggs in a bubble nest. The complex sexual behavior and nest-building of fish belonging to the Anabantoidei species has been described in detail [2,5,6,7]. In natural habitats according to water space availability for a mature male, the male becomes territorial, builds a bubble nest, and protects it from other males. The female swims under the nest following sexual behavior and spawns eggs into the bubble nest (Figure 1). The male becomes territorial during the reproduction cycle as is common in other fishes, e.g., *Tilapia zilli* [8].

In blue gourami as a model fish of hormone control reproduction for Labyrinithici fish, the sexual behavior of the male [11] and pheromones [12,13] affect the gonadotropic axis—the brain-pituitary-gonadal (BHPG) axis [14] and the somatotropic axis [15]—the brain-pituitary-liver and body (BPLB) axis. Pheromones involved in reproduction are also described in other fish, [16] and directly affect BPG.

These axes are very complex and they control the oocytes phase from vitellogenesis to maturation, ovulation, and spawning [1,10,11] (Figure 1H–J). The function of the bubble nest is to supply oxygen to the eggs and larvae in the water in places where the concentration is very low [1,14].

## 2. Environmental, Internal, and Sexual Behavior Factors Affecting the Female Brain

The brain of Anabantoidei fish contains neurons secreting hormones that are involved in growth and reproduction control and the interactions between them, which is very complex and is summarized in Figure 2. The environment, pheromones, sexual behavior, and hormones are involved in oogenesis and the interactions between the two axes of blue gourami, BPG and BPLB, as is presented in Figure 2 and supported by relatively many studies [9,10,11,12,13,17,18,19,20,21,22,23,24,25,26,27,28,29,30,31,32,33,34,35,36,37,38,39,40,41,42,43,44,45,46,47,48,49].

Environmental conditions such as water temperature [53], sexual behavior [11], and pheromones [12,13] affect the brain within the neuroendocrine system in Labyrinithici fish, as described in detail for other teleosts [54]. The brain is the main organ that controls reproduction along the hypothalamus-pituitary-gonadal (HPG) axis [14]. Oogenesis and spermatogenesis in fish like in other vertebrates are a complex mechanism; many different processes are involved in the brain and it is difficult to differentiate between them (Figure 3). However, the difference from other fish is the male behavior effect on oogenesis and the change from vitellogenesis to maturation (Figure 1).

The transfer of oocytes from pre-vitellogenesis to vitellogenesis to maturation is controlled by the BPG axis. The brain transcriptome in female blue gourami is changed dramatically during puberty [55]. In the brain of blue gourami, 34,368 unique transcripts were identified, 23,710 of which were similar to other species. In the brain of female blue gourami, different genes were upregulated in juveniles (pre-vitellogenesis) compared to adults (vitellogenesis) (Figure 3, Table 1).

The expression of those genes is different in the brain of juvenile females found in pre-vitellogenesis than in the brain of mature fishes whose ovaries are in vitellogenesis. The advances of the present indicate that many genes from the transcriptome analysis in Table 1 are expressed differently in the brain at PVTL and HVTL. The functions of the genes are not always known.

More specifically, it seems that that relatively many neurons in the brain affect Kisspeptin (Kiss), which controls GnRHs (GnRH1 and 3) and HPG, as described in other teleost fish [54]. My hypothesis is that, in blue gourami as in other Labyrinithici fish, the environment, pheromones, and sexual behavior of males affect Kisspeptin, which controls GnRH 1 and 3. Much of our data support this hypothesis, but more direct studies are required to further support it.

Degani and colleagues [18] described the mRNA level of Kisspeptin2 (Kiss2) and Kiss receptors (GPR54 or Kiss2r, Kiss1r) in the brain of blue gourami [1] (Figure 4, Figure 5 and Figure 6). These hormones are also described in other teleost fish, e.g., zebrafish (*Danio rerio*) [56,57], lamprey (*Petromyzon marinus*) [56], medaka (*Oryzias latipes*) [58], gold fish (*Carassius auratus*) [59] and sea bass (*Morone saxatilis*) [60].

## 3. The Hormones Involved in Reproduction and Growth in the Brain of Blue Gourami

It is generally accepted that, in fish, the Kiss controlling the neuropeptides released by these nerve endings are gonadotrophin-releasing hormones (GnRH) and dopamine [61]. In the female blue gourami model fish belonging to the Anabantoidei fish, three GnRHs (GnRH1, 2 and 3) have been studied in detail [10,17].

Based on the results [17] (Figure 4, Figure 5 and Figure 6) of three GnRHs cloning and expression during growth and pubescence, and Kiss2 expression and its receptors, it is suggested that the effect on Kiss2 affects GnRH1, which controls and regulates vitellogenesis in blue gourami [18,25], and both GnRH1 and 2 control maturation ovulation and spawning.

Although it is difficult to distinguish between the function of GnRH1 and 3 in females (GnRH2 does not directly affect oogenesis), apparently GnRH1 has a greater effect during pubescence and on follicular stimulated hormone (FSH) and GnRH3, as well as on the luteinize hormone (LH) [17,20], which controls oocytes maturation (OM) and ovulation (OV) (Figure 6). It seems that GnRH3 and PACAP [17], which dramatically alter gene expression during oocyte maturation in blue gourami, control this process (Figure 6). Both OM and OV are affected by male sexual behavior and male pheromones (Figure 1) (reviewed in [1,11]).

The effects of both GnRH1 and GnRH3 have a very similar response to the hormones administrated in blue gourami and the sequences are very similar [17] (Figure 6). Apparently, the growth that takes place during the process of pubescence is concomitant with vitellogenesis (which is not dependent on male pheromones and behavior) [11,12]. Kiss2, which controls GnRH1 and affects FSH and E_2_ [25] secretion, controls vitellogenesis [18] (Figure 6) and 17, 20P during maturation.

## 4. Pituitary Hormones Control Reproduction in Female Blue Gourami

The brain controls the pituitary gland hormones in blue gourami as in other teleost fish and is the main organ that is involved in effects on the gonadotropic axis—The brain-pituitary-gonadal (BHPG) axis [14,54]. However, the somatotropic axis—the brain-pituitary-liver and body (BPLB) axis—is also important, and the interactions between the two axes are very complex.

Detailed studies have been carried out on hormones in the pituitary of blue gourami using histological methods based on light and electron microscopy [49], as well as molecular methods [9,10,30,39,50], which might help in understanding the control of oogenesis in this specific group in which blue gourami is represented. FSH and LH mRNA levels are very high during advanced vitellogenesis and maturation, and both are involved in two different stages [39] (Figure 6 and Figure 7). Both gonadotropins, which are affected by steroids [23,24], control oogenesis (Figure 7). Blue gourami is a group of multi-spawning fish. Sometimes, one group of oocytes arrives at maturation and another group is found at the vitellogenesis stage (Figure 1). This is the proposed explanation for the high transcription of both FSH and LH at advanced vitellogenesis and maturation (Figure 7) [10,39].

The cytochrome P450 aromatase gene (CYP19) seems to be involved directly and indirectly in the oogenesis of blue gourami [37]. During sterogeensis in the ovary, the synthesis of estrogens from androgens is catalyzed by the enzyme cytochrome P450 aromatase (CYP19) in vertebrates. In blue gourami brain, CYP19 was found (bgCYP19b) in which mRNA expression was very low in oocytes found in vitellogenesis but increased dramatically in oocytes in maturation, and gonadal CYP19 (bgCYP19a) was relatively high in the gonads. The mRNA bgCYP19a expression was measured during various stages of oogenesis. The aromatase gene mRNAb levels during oogenesis were significantly lower in females during the vitellogenesis stage compared to females during maturation [37] (Figure 8), suggesting that this hormone is involved in the sexual behavior of female blue gourami and not in the control of oogenesis [37].

Both GH and IGF1 are involved in oogenesis, as well as prolactin (PRL) [35,50,62] (Figure 9). The levels of GH and PL mRNA genes were found to be relatively constant during growth and oogenesis, and no significant differences were found (Figure 9) [35,50,62]. These hormones are integral hormones that affect many biological processes, including reproduction.

Growth hormone (GH) and insulin-like growth factor 1 (IGF-1) are central hormones involved in somatic growth (somatotropic axis) [29], as well as in reproductive functions, and they are under multifactorial regulation by pituitary adenylate cyclase-activating polypeptide (PACAP) and its related peptide (PRP) [20]. These peptides coordinate as part of the complex regulatory mechanisms of reproduction at the brain-pituitary level; therefore, understanding the complex hormonal interactions regulating this axis is crucial, particularly in fish as a simple model system of female blue gourami [17,19,20]. All the various neuron secretory hormones in the brain with the gonadotropic and somatotropic axes control oogenesis in Anabantoidei fish, Kiss, GnRH1, GnRH2, PACAP, and PRP [17,18,20,55].

In summary, all of the complex interactions between hormones in the BPG and BPLB axes of the quality model of female blue gourami are proposed (Figure 10).

In conclusions, this paper showed the complex various factors involved in the interaction between environment pheromones and hormones that involve reproduction Anabantiform order and have not received as much attention as other fish. This unique fish group confirms continued research and highlights various aspects of control reproduction that have a general contribution to understanding the complexity of breeding control.

## Figures and Tables

**Figure 1 biology-09-00109-f001:**
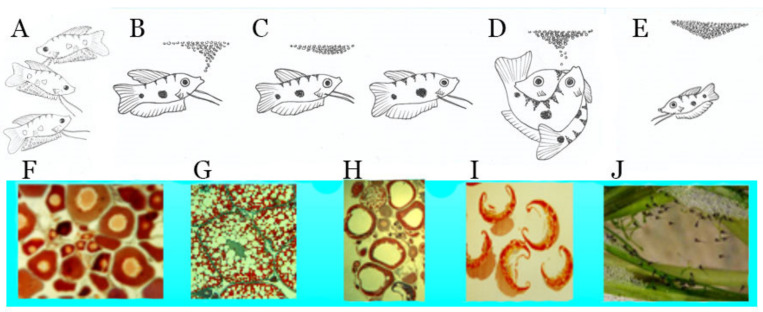
The relationship between sexual behavior, oogenesis and nursing in blue gourami [1]. (**A**) High density—no reproduction occurs. (**B**) The male builds a bubble nest. (**C**) Sexual behavior under the nest. (**D**) The male wraps his body around the female and the female spawns eggs that will be deposited in the nest. (**E**) The bubble nest with eggs in it. (**F**) Oocytes in pre-vitellogenesis. (**G**) Oocytes in vitellogenesis. (**H**) Oocytes in maturation. (**I**) Oocytes in ovulation. (**J**) Fry hatches in the nest [9,10].

**Figure 2 biology-09-00109-f002:**
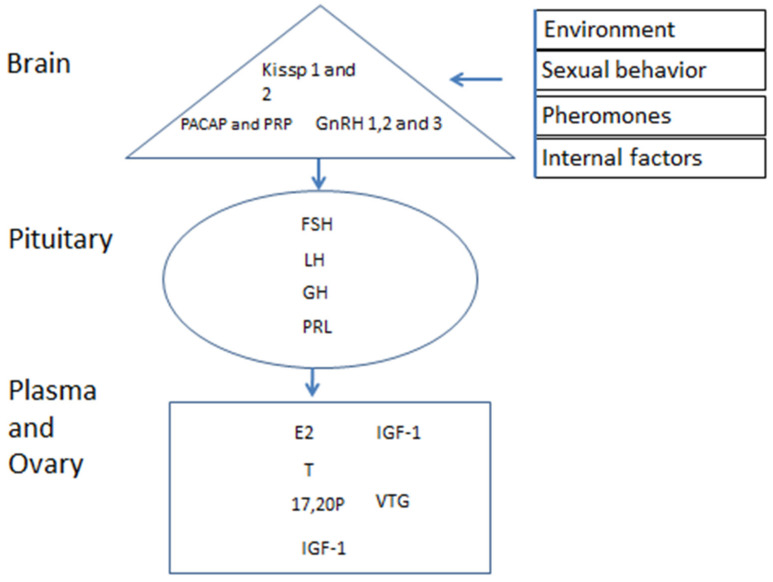
Factors affecting the hormones of female reproduction. The environments [19,32], sexual behavior [11] and pheromones [12], Kisspeptin (Kiss 1 and 2) [18], gonadotropin-releasing hormone (GnRH1 and 3) [20], follicle-stimulating hormone (FSH), luteinizing hormone (LH) [39,42], pituitary adenylate cyclase-activating polypeptide (PACAP) and its related peptide (PRP) [20], growth hormone (GH) [50], prolactin (PRL) [35]. The FSH and LH act on the ovary to synthesize steroids, 17β-estradiol (E2) [51,52], testosterone (T) [51,52], and 17α,20β- dihydroxy-4-pregnen-3-one (17,20P) [51,52], and in the liver, synthesis vitellogenin (VTG) [45] and insulin-like growth factor 1 (IGF1) [29].

**Figure 3 biology-09-00109-f003:**
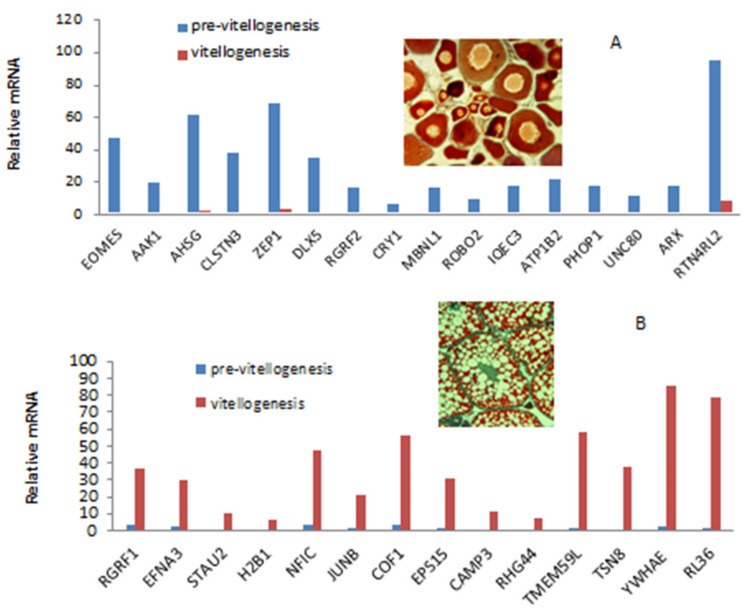
mRNA levels (based on RNA-Seq) of transcripts associated with known genes (Table 1) representing the differences in edgeR analysis (*p* < 0.001) between pre-vitellogenesis (**A**) and vitellogenesis (**B**) in female blue gourami brains [55].

**Figure 4 biology-09-00109-f004:**
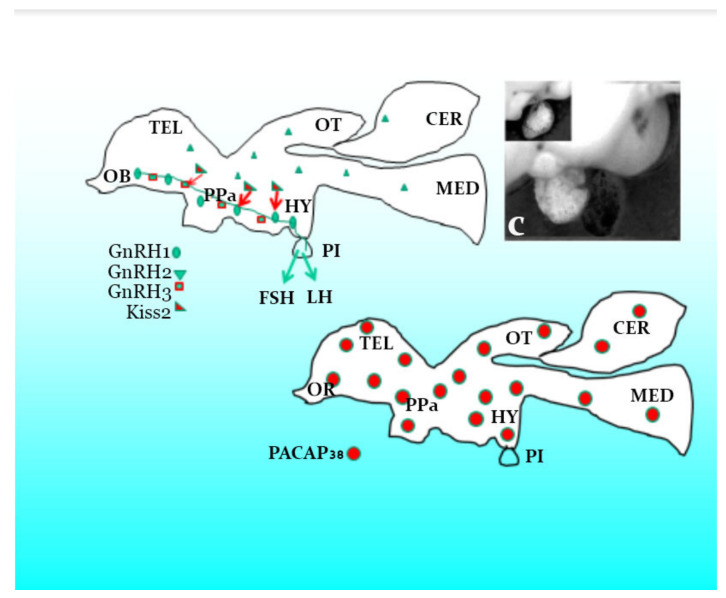
Proposed sketch of hormones secretion from the brain and pituitary of female blue gourami s based on gene expression in various studies [10,17,18,20,30,31,34,36,55]. OB—Olfactory bulb, OT—Optic tectum, CER—Carpus cerebelii, PPa—Nucleus preopticus parvicellularis posterioris, PI—Pituitary gland, HY—Hypothalamus, Kiss2—Kisspeptin 2, GnRH1, 2 and 3—Gonadotropin-releasing hormone 1, 2 and 3, PACAP38—Pituitary adenylate cyclase activating polypeptide, FSH—Follicle-stimulating hormone, LH—Luteinizing hormone.

**Figure 5 biology-09-00109-f005:**
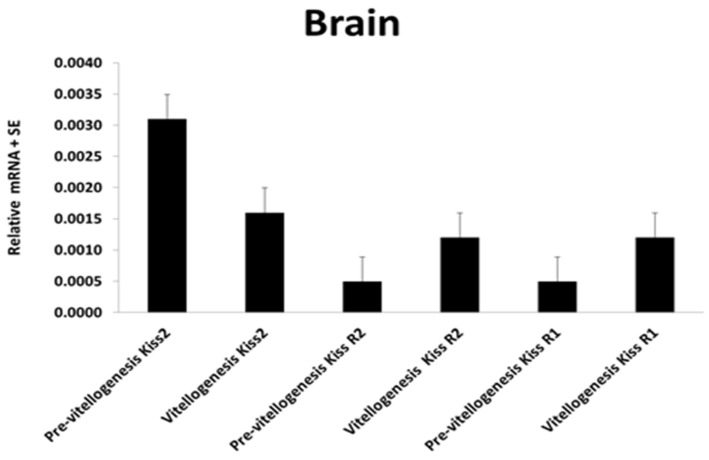
Comparing the gene transcriptions of Kiss2, KissR1, and KissR2 in the brain of female blue gourami during oogenesis [18].

**Figure 6 biology-09-00109-f006:**
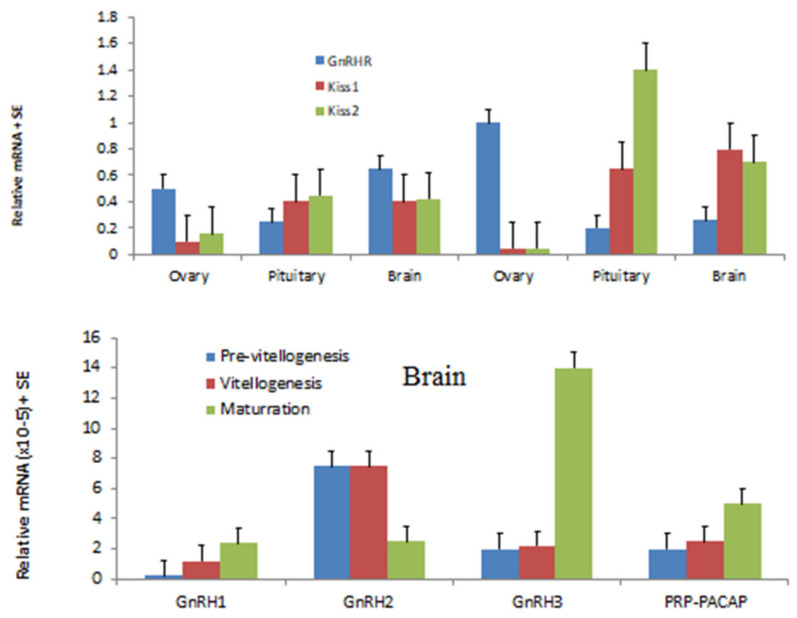
The relationship between the transcription of Kisspeptin (Kiss) 1 and 2 in the brain and the mRNA level of gonadotropin-releasing hormone (GnRH1, 2, 3), pituitary adenylate cyclase-activating polypeptide (PACAP) in various tissues of female blue gourami is presented [17,18,20,31,55].

**Figure 7 biology-09-00109-f007:**
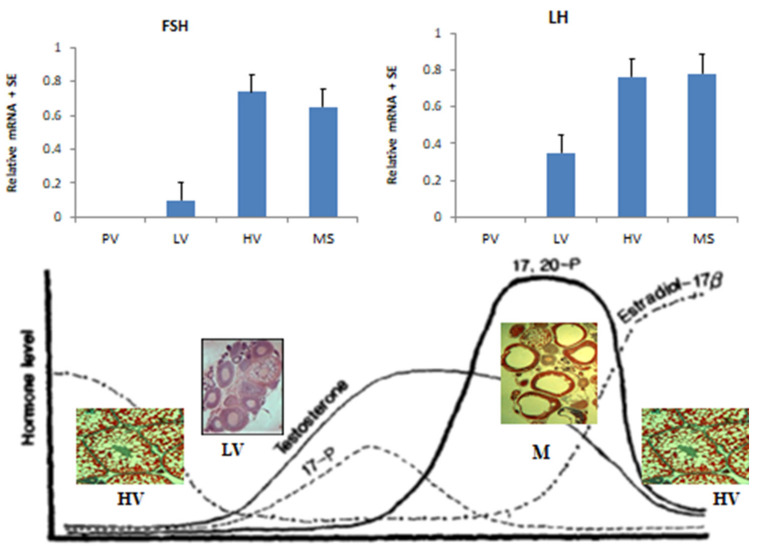
Proposed quality model showing the relationship between gonadotropins levels and steroids during the gonadal cycle, low vitellogenesis (LV), high vitellogenesis (HV), and maturation (MS). The steroids are 17β-estradiol (E2), testosterone (T), 17α,20β- dihydroxy-4-pregnen-3-one (17,20-P) [10,23,24,39].

**Figure 8 biology-09-00109-f008:**
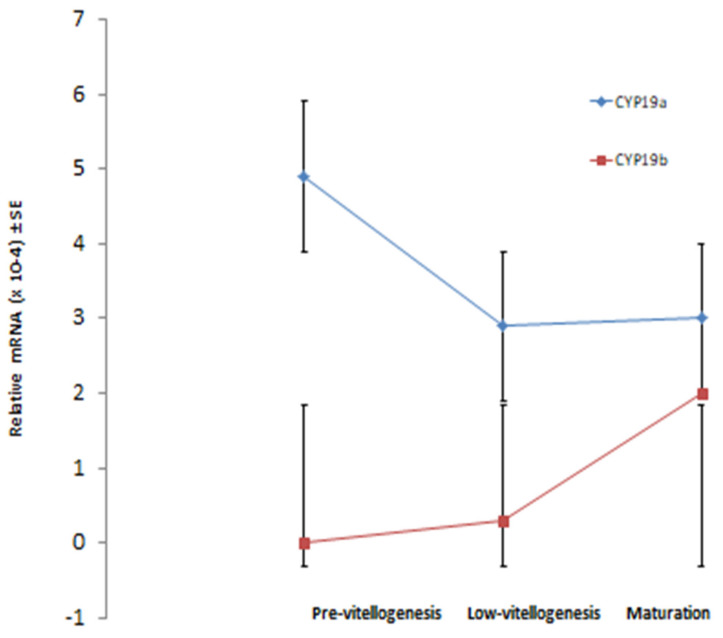
The transcription of ovary aromatase (CYP19a) and brain aromatase (CYP19a) at various stages of oocytes in vitellogenesis and maturation of female blue gourami [37].

**Figure 9 biology-09-00109-f009:**
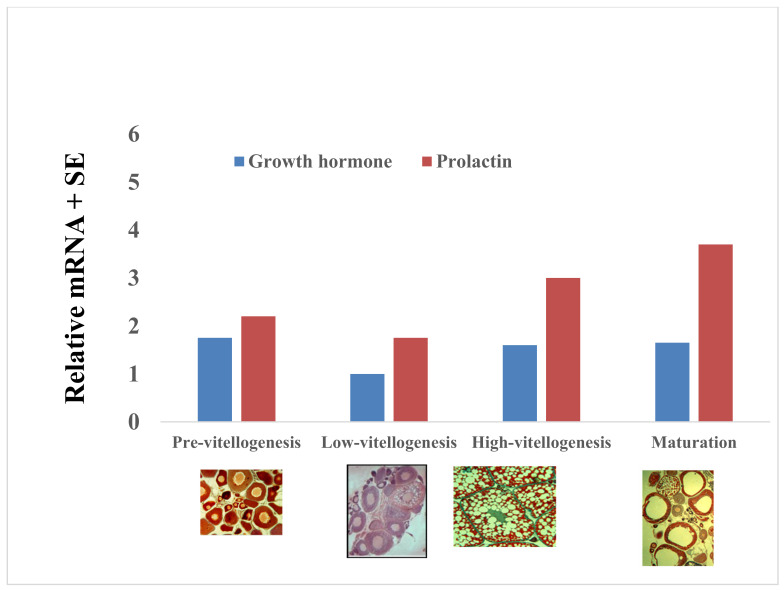
Variations in GH and prolactin mRNA levels during the gonadal cycle (X 8.5 × 10^−6^): pre-vitellogenesis (PV), low vitellogenesis (LV), high vitellogenesis (HV), and maturation (MS). Each histogram represents the average of five independent measurements (mean ± SE) [35,50].

**Figure 10 biology-09-00109-f010:**
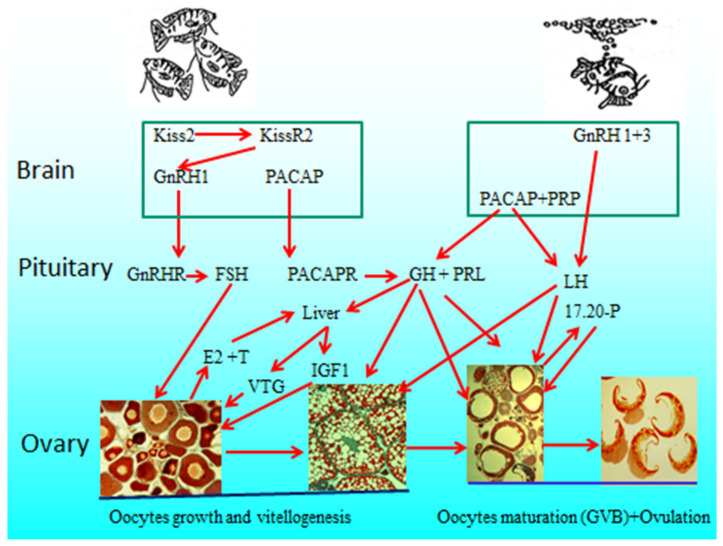
The interactions between the two axes, BPG and BPLB, of female blue gourami control oogenesis based on the review and unpublished data [9,10,11,12,13,17,18,19,20,21,22,23,24,25,26,27,28,29,30,31,32,33,34,35,36,37,38,39,40,41,42,43,44,45,46,47,48,49]. Gonadotropin-releasing hormone 1 and 3 (GnRH1 and 3), follicle-stimulating hormone (FSH), luteinizing hormone (LH), pituitary adenylate cyclase-activating polypeptide (PACAP) and its related peptide (PRP), growth hormone (GH), prolactin (PRL), 17β-estradiol (E2), testosterone (T), 17α,20β- dihydroxy-4-pregnen-3-one (17,20P), vitellogenin (VTG), insulin-like growth factor 1 (IGF1).

**Table 1 biology-09-00109-t001:** In the whole-brain mRNA level list of gene names and functions, changes are different in PVTL and HVTL, which supports some connections and functions of those genes in oogenesis [55]. All of the information is based on GenBank Functon.

Acronyms Gene ID	Full Name of the Gene	GenBank Function
Pre-Vitellogenesis		
CERS6	*Ceramide synthase 6*	DNA binding Dihydroceramide synthase. Catalyzes the acylation of sphingosine to form dihydroceramide.
CERS5	*Ceramide synthase 5*	
AAK1	*AP2 associated kinase 1*	Regulates clathrin-mediated endocytosis by phosphorylating the AP2M1/mu2 subunit of the adaptor protein complex 2 (AP-2), which ensures high affinity binding of AP-2 to cargo membrane proteins during the initial stages of endocytosis.
AHSG	*(FETUA) alpha 2-HS glycoprotein*	Probably involved in differentiation.
CLSTN3	*Calsyntenin 3*	May modulate calcium-mediated postsynaptic signals. Complex formation with APBA2 and APP, stabilizes APP metabolism and enhances APBA2-mediated suppression of beta-APP40 secretion due to the retardation of intracellular APP maturation.
ZEP1	*Zeaxanthin epoxidase*	Converts zeaxanthin into antheraxanthin and subsequently violaxanthin.
DLX5	*Distal-less homeobox 5*	Transcriptional factor involved in bone development. Acts as an immediate early BMP-responsive transcriptional activator essential for osteoblast differentiation.
RGRF2	*Ras guanine nucleotide exchange factor 2*	Functions as a calcium-regulated nucleotide exchange factor activating both Ras and rac1 through the exchange of bound GDP for GTP. May function in synaptic plasticity.
CRY1	*Cryptochrome-1*	Transcriptional repressor that forms a core component of the circadian clock.
MBNL1	*Muscle blind like splicing regulator 1*	Negative regulation of axon extension involved in axon guidance. Metal ion binding.
ROBO2	*Roundabout guidance receptor 2*	Anterior/posterior axon guidance. Central nervous system projection neuron axonogenesis.
IQEC3	*IQ motif and Sec7 domain 3*	Acts as a guanine nucleotide exchange factor (GEF) for ARF1.
ATP1B2	*ATPase Na+/K+ transporting subunit Beta 2*	This is the non-catalytic component of the active enzyme, which catalyzes the hydrolysis of ATP coupled with the exchange of Na^+^ and K^+^ ions across the plasma membrane. The exact function of the beta-2 subunit is not known.
PHOP1	*Probable phosphatase phospho1*	Probable phosphatase, involved in bone mineralization.
UNC80	*Unc-80 homolog, NALCN activator*	Component of the NALCN sodium channel complex required for channel regulation. UNC80 is essential for NALCN sensitivity to extracellular calcium.
ARX	*Aristaless related homeobox*	Appears to be indispensable for central nervous system development. May play a role in the neuronal differentiation of the ganglionic eminence and ventral thalamus. May also be involved in axonal guidance in the floor plate.
Rtn4rl2	*Reticulon-4 receptor-like 2*	Cell surface receptor. Plays a functionally redundant role in postnatal brain development and in regulating axon regeneration in the adult central nervous system. Contributes to normal axon migration across the brain midline and normal formation of the corpus callosum. Protects motoneurons against apoptosis.
**Vitellogenesis**		
RGRF1	*Ras Protein Specific Guanine Nucleotide Releasing Factor 1*	Promotes the exchange of Ras-bound GDP by GTP.
EFNA3	*Ephrin A3*	Binds promiscuously Eph receptors residing on adjacent cells, leading to contact-dependent bidirectional signaling into neighboring cells.
STAU2	*Staufen double-stranded RNA binding protein 2*	RNA binding.
H2B1	*Histone H2B-like*	DNA binding, protein heterodimerization activity.
NFIC	*Nuclear factor I C*	Recognizes and binds the palindromic sequence 5′-TTGGCNNNNNGCCAA-3′ present in viral and cellular promoters and in the origin of replication of adenovirus type 2. These proteins are individually capable of activating transcription and replication.
JUNB	*JunB proto-oncogene, AP-1 transcription factor subunit*	RNA polymerase II core promoter proximal region sequence-specific DNA binding. RNA polymerase II transcription factor activity, sequence-specific DNA binding. Transcription coactivator activity, transcription factor binding.
COF1	*Cofilin*	Binds to F-actin and exhibits pH-sensitive F-actin depolymerizing activity. Regulates actin cytoskeleton dynamics. Important for normal progress through mitosis and normal cytokinesis.
EPS15	*Epidermal growth factor receptor pathway substrate 15*	Involved in cell growth regulation. May be involved in the regulation of mitogenic signals and control of cell proliferation. Involved in the internalization of ligand-inducible receptors of the receptor tyrosine kinase (RTK) type, in particular EGFR.
CAMP3	*Calmodulin-regulated spectrin-associated protein 3*	Microtubule minus-end binding protein that acts as a regulator of non-centrosomal microtubule dynamics and organization. Specifically required for the biogenesis and maintenance of zonula adherens by anchoring the minus-end of microtubules to zonula adherens and by recruiting the kinesin KIFC3 to those junctional sites.
RHG44	*Rho GTPase-activating protein 44-like*	GTPase-activating protein (GAP) that stimulates the GTPase activity of Rho-type GTPases. Thereby, controls Rho-type GTPases cycling between their active GTP-bound and inactive GDP-bound states.
TMEM59L	*Transmembrane protein 59 like*	Modulates the O-glycosylation and complex N-glycosylation steps occurring during the Golgi maturation of APP. Inhibits APP transport to the cell surface and further shedding.
TSN8	*Tetraspanin-8*	Integrin binding
YWHAE (1433E)	*Tyrosine 3-monooxygenase/tryptophan 5-monooxygenase activation protein epsilon*	Monooxygenase activity. Protein domain specific binding.
RL36	*Ribosomal protein 36 60S large ribosomal subunit*	Component of the large ribosomal subunit.

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
