# Peer review of "Brain Control Reproduction by the Endocrine System of Female Blue Gourami (Trichogaster trichopterus)"

_biology, 2020, doi:10.3390/biology9050109_

Round 1

Reviewer 1 Report

GENERAL COMMENTS

I have some concerns with this paper, the most important one relies on what the author call it as a “mini review”. Therefore, as far as I understand this is not a complete review as it should be, but then a short one, covering only a part of the subject to be analyzed, in particular the works of the author. That can be seen by the huge amount of references of the author (c. 80%), either as 1st or as co-author. That what troubles me the most, that this has not a full coverage (as the author says, it is a “mini review”). It seems the author is making a resume of his own work. There are also several issues, apart from a lack of structure (some items are numbered, other not; there is hardly a sequence) which denote lack of attention throughout the manuscript, as the format of references at the final is a mess! There is hardly no format and this also happens on the text with a lot of blank spaces. The quality of figures is very low, it seems they were copied-pasted from the works of the author and placed here without any care for resolution. No conclusions and future lines of research to be pursue. In summary, I think the authors should re-think the coverage of his mini review and embrace a full coverage of the theme, while paying attention to the formats and quality of outputs of the study.

SPECIFIC COMMENT

Figures are generally in poor quality and should be re-drawn in a better quality. It seems they were taken from previous articles of the author according to the citation on the captions.

References – wrongly inserted in the text: they must be numbered in order of appearance in the text. They are also wrongly formatted in the list, without ant careful formatting (see for example the case, among many of lines 281-283).

Formatting of the text – there are several blanks spaces throughout the text (particularly between cited references), please check them accordingly.

L6 – Provide species scientific name at first common name citation.

L8 – What do you mean by “(Labyrinth)” in this context?

L12 and throughout the manuscript – I don’t know what you understand by “mini review”. Is this a review or not? Did you skip some of the documents available for this manuscript? Please clarify and remove the “mini” of the text. If this is not a full review it should not be stated as a review. Otherwise, change the title and focus of the manuscript.

L23 – I think a final sentence of applicability and broader scope of the findings should be provided to increase interest.

L25 – Again, provide species scientific name upon first citation in the text.

L35 – Remove space.

L57 – Titles and subtitles should be numbered as you did on the Intro (L24).

L72-83 – Please avoid such long figure captions.

L91- Not clear this sentence.

Figure 3

Table 1 – Here you should provide further details of what you are showing on the table, not just Genes (of what?) expressed differentially in PVTL/HVTL  (?)(as in Fig. 3) and their putative functions. Avoid citing other figures/tables in the captions of other figures/tables.

L104-106 – This sentence should be clearer, as it is difficult to understand what authors want to say. There seems to be some speculation without a support for the statements herein.

L111- What data? Any reference?

Figure 4 – Once again, too low quality. It seems it was taken from elsewhere and pasted here.

L137 – It seems this article dating back of 2017 is still “in press”…

L155-156 – Revise the sentence, it seems a verb is missing here. Check “and 17, 20P during…”

What are the conclusions of this study? What are the future lines of research?

Author Response

General Comments

  1. The paper is a “mini review” because it covers only the reproduction of the Anabantoidei suborder of anabantiform fish, which is a relatively large systematic group having over 30,000 species. The reproduction control of systems in this group has a high variability and is covered by all subjects of the review. I changed the name of the “mini review” to a review on brain control reproduction of the Anabantoidei suborder.
  2. I must admit that I had been involved in many of the studies, the main one being the current study. However, other scientists are also involved with the review and contributed to it. I have added more references in addition to the studies that I have conducted.
  3. I improved the quality of the figures and redone some of them completely.
  4. I added conclusions and future lines of research to be pursued.

Referee 1 (yellow)

  1. I redid Figures 1 and 4 in better quality.
  2. I changed the references in the text by number.
  3. I checked the formatting of the text and removed blank spaces throughout the (particularly between cited references).
  4. L40 – I provided the species scientific name at the first common name citation of blue gourami (Trichogaster trichopterus).
  5. I removed the word “mini” throughout the text.
  6. I removed spaces throughout the text.
  7. I numbered the titles according to the Introduction.
  8. I deleted the line: Brain involvement in reproduction in Labyrinithici fish of female blue gourami model fish
  9. I shortened the Figure 2 legend.
  10. I clarified the sentence in Line 91.
  11. I added more explanations about the importance of the dada in all of the tables.
  12. L102-103 – I clarified this sentence.
  13. L137 – It seems this article dating back to 2017 is still “in press.”
  14. L155-156 – The sentence was revised
  15. I now included the conclusions of this study and the future lines of research.

Reviewer 2 Report

In his paper entitled “Brain control reproduction by the endocrine system of female blue gourami (Trichogaster trichopterus)” Dr. Degani summarizes available information on the hormones that control the reproduction system of female blue gourami, a fish characterized by the labyrinth, an organ that allows respiration of atmospheric oxygen.

The paper is of potential interest for the Readers of Biology. However, some points should be addressed before acceptance:

  1. Citations given in the paper are mainly from the Author and his collaborators (47 out of a total of 60); although the Author is certainly an expert in the field, to refer also to a more general context should improve the review;
  2. In Table 1, references should be added in columns 1 and 3;
  3. Fig. 4 is of poor quality, and not clearly explained in the corresponding legend; it should be improved or deleted at all;
  4. Legend to Fig. 6 seems to refer only to the upper part of the figure, while no reference is done to the lower one; in addition, in the lower part “maturration” should be corrected;
  5. Finally, a general revision of English is suggested.

Author Response

General Comments

  1. The paper is a “mini review” because it covers only the reproduction of the Anabantoidei suborder of anabantiform fish, which is a relatively large systematic group having over 30,000 species. The reproduction control of systems in this group has a high variability and is covered by all subjects of the review. I changed the name of the “mini review” to a review on brain control reproduction of the Anabantoidei suborder.
  2. I must admit that I had been involved in many of the studies, the main one being the current study. However, other scientists are also involved with the review and contributed to it. I have added more references in addition to the studies that I have conducted.
  3. I improved the quality of the figures and redone some of them completely.
  4. I added conclusions and future lines of research to be pursued.

    Review 2 (green)

    1. I added more papers on the reproduction of other fish.
    2. Table 1 is based on GenBank Function.
    3.    Figure 4 was redone.
    1. I corrected the Figure 6 legend.
    2. I gave the paper English editor to correct my language before resubmitting it.

Round 2

Reviewer 1 Report

I thank the authors for addressing (and correcting accordingly, specially the mini-review issue) my concerns regarding the previous submission. Only a couple of minor issues to dealt with:

L120-123 (Table 1) – This caption does not agree with the table, which is a (long) list of gene names and functions. Anyway, with reference to the caption, consider the following: i) “form” or “from”? ii) avoid using “gene” twice in the same sentence, iii) avoid referring to other figures or tables in a caption of a figure/table (as you have here, calling to Fig. 3).

L183-186 – Avoid the term “In this figure” in a caption of a figure. Suggest something like “The relationship between the…”-

L253 – Replace the x by 10.

L272-273 – Please re-write this sentence, as it seems some words are duplicated and an error (macar?): “are detail more detail study' but this knowledge may have the A first step and opening a gate to the macar promotes the breeding of this group of fish.”.

Reviewer 2 Report

The Authors replied to most of criticism raised by this Reviewer. However, the paper still requires minor modifications:

  1. In the title, is the word “control” used as a verb or as a noon? If it is intended as a verb, then it should be better to write: “Brain controls reproduction of female blue gourami (Trichogaster trichopterus) by (or through) the endocrine system”; If it is intended as a noon, perhaps it should be better to write: “Brain control of reproduction in the female … is mediated by the endocrine system”;
  1. Abstract: in the new sentence “This paper is summarizes the complex various factors involved…”, “is” should be deleted, and I suggest to write “the various and complex factors”;
  2. Abstract, line 26: is “vitellogenesis” intended as “vitellogenin”? Please note that, if the meaning here is vitellogenin, the acronym (VTL) cannot be different from the one used in the legend to Fig. 2 (VTG);
  3. In the legend to Figure 2, I suggest to change the sentence: “The FSH and LH act on the ovary to synthesize steroids, 17β-estradiol (E2) [51,52], testosterone (T) [51,52], and 17α,20β- dihydroxy-4-pregnen-3-one (17,20P) [51,52], and in the liver, synthesis vitellogenin (VTG) [45] and insulin-like growth factor 1 (IGF1) [29]” into “FSH and LH act on the ovary where they induce the synthesis of steroids, 17β-estradiol (E2) [51,52], testosterone (T) [51,52], and 17α,20β- dihydroxy-4-pregnen-3-one (17,20P) [51,52], and on the liver, where they induce the synthesis of vitellogenin (VTG) [45] and insulin-like growth factor 1 (IGF1) [29];
  4. Lines 120-122: Legend to Table 1, the sentence “Form whole-brain transcriptome the genes change during vitellogenesis in the brain differentially in some genes that represented different PVTL and HVTL (as in Fig. 3), support some connections and functions of those genes in oogenesis” (in which “Form” should be “from”, I suppose) might be changed to: “List of genes that show differential activation during vitellogenesis, on the basis of whole-brain transcriptome analysis (see also Figure 3). This analysis supports some connections and functions of these genes in oogenesis. Etc.
  1. In the same legend to Table 1: please specify the meaning of PVTL and HVTL in extensor;
  2. Lines 127-130: these sentences should be improved as long as it concerns English language;
  3. Line 134: delete one “that”;
  4. Line 270: the sentence ”… between environment pheromones and hormones that involvement of reproduction…” should be changed into “…between environment, pheromones, and hormones that are involved in the reproduction of…”.
